# OpenReview forum: "Dual Intention Escape: Jailbreak Attack against Large Language Models"
_ACM.org/TheWebConf/2025/Conference — WWW 2025 Oral_

### Official Review · Reviewer_Em4M · 2024-11-07

**Novelty:** 5
**Technical Quality:** 5

**Review:**

This paper presents the Dual Intention Escape (DIE) jailbreak attack framework designed to exploit psychological misjudgment mechanisms in large language models (LLMs). The DIE framework combines Intention-anchored Malicious Concealment (IMC) for evading detection and Intention-reinforced Malicious Inducement (IMI) for increasing harmfulness. Experimental results show that DIE significantly outperforms prior methods in success rate and toxicity across models, particularly under black-box settings.

Strengths:

The paper addresses a highly relevant and popular research area, focusing on the vulnerabilities of large language models.

Innovative application of psychological mechanisms (anchoring and availability bias) to enhance jailbreak efficacy.

Experimental results demonstrating superior attack performance on widely-used LLMs like GPT-3.5 and LLaMa models.


Weaknesses:

Limited exploration of post-hoc defense mechanisms. The paper primarily addresses pre-hoc defenses but lacks in-depth exploration of post-hoc or adaptive defense mechanisms that could mitigate the DIE attack’s effects. Including such analysis would strengthen the practical applicability of the findings.

**Questions:**

See above.

**Reviewer Confidence:**

3: The reviewer is confident but not certain that the evaluation is correct

**Scope:**

3: The work is somewhat relevant to the Web and to the track, and is of narrow interest to a sub-community

---

### Official Review · Reviewer_gKBc · 2024-11-25

**Novelty:** 5
**Technical Quality:** 4

**Review:**

This paper addresses a critical and emerging issue in the domain of LLM security: the vulnerability of these models to adversarial attacks. It introduces the Dual Intention Escape (DIE) framework, which combines psychological insights with technical innovations to improve both the penetrability and toxicity of jailbreak attacks. This dual approach, leveraging the anchoring effect and availability bias, demonstrates originality and contributes to advancing the understanding of LLM vulnerabilities. The methodological rigor, including the design of the Intention-Anchored Malicious Concealment (IMC) and Intention-Reinforced Malicious Inducement (IMI) modules, is noteworthy, and the experimental results show clear improvements over existing methods in terms of attack success rates and response toxicity.

The paper effectively describes complex ideas, such as recursive decomposition and contrary intention nesting, while maintaining accessibility for readers familiar with adversarial machine learning. However, certain sections, such as the mathematical formalizations and algorithmic descriptions, could benefit from illustrative examples to ensure broader comprehension. Additionally, the figures and tables are informative but could be expanded to provide more visual insights into the attack mechanisms and outcomes.

The incorporation of psychological principles to design more effective jailbreak attacks represents a novel perspective in the field. The authors provide a valuable foundation for future research into both adversarial attacks and mitigation strategies, emphasizing the dual role of such research in exposing vulnerabilities and guiding improvements in AI alignment. However, the ethical concerns surrounding the development and dissemination of more potent attack strategies are only briefly addressed, leaving room for a more nuanced discussion about the implications and potential misuse of this work.

The experimental results are convincing, showing significant improvements over comparable methods, particularly in attack success rates and toxicity amplification. The limitations, as acknowledged by the authors, include challenges with post-hoc defense strategies and the difficulty of further enhancing attack toxicity. Additionally, the ethical framework could be strengthened to balance the technical achievements with societal concerns more effectively.

**Questions:**

1.The paper briefly addresses ethical concerns but does not elaborate on safeguards against potential misuse of the proposed DIE framework. Could you provide more details on the ethical guidelines or limitations placed on the dissemination of your findings?

2.The paper uses ASR-R and ASR-G as evaluation metrics. While ASR-G incorporates toxicity, it still relies on GPT-3.5-turbo as an evaluator. How do you address potential biases in this evaluation model, and are there alternative methods to validate toxicity?

3.The IMC and IMI modules involve recursive decomposition and multi-level inducement, which could be computationally expensive. Did you measure the time and resource costs of generating adversarial prompts, and how feasible is the framework for large-scale deployment?

4.The framework effectively bypasses existing defenses in LLMs. Based on your findings, what specific types of defensive strategies would you recommend to counteract the DIE attack?

5.The recursive depth and the number of sub-intentions are key hyperparameters in your method. How sensitive are the attack success rates and toxicity levels to variations in these parameters?

**Reviewer Confidence:**

4: The reviewer is certain that the evaluation is correct and very familiar with the relevant literature

**Scope:**

4: The work is relevant to the Web and to the track, and is of broad interest to the community

---

### Official Review · Reviewer_GYx9 · 2024-11-28

**Novelty:** 5
**Technical Quality:** 5

**Review:**

This paper proposes a novel jailbreak attack method called DIE, inspired by the psychology of human misjudgment. The method employs an IMC module to construct benign anchor prompts that guide large language models (LLMs) to produce harmless outputs related to the prompts. Subsequently, the IMI module crafts malice-correlated prompts to induce harmful outputs. Through multi-turn dialogues, the DIE method jailbreaks LLMs.

### Pros:

1. The paper features high-quality illustrations that are easy to read and well-organized.
2. The method is based on insights from the psychology of human misjudgment, which is an interesting and relevant perspective.

### Cons:

1. The paper lacks innovation. The design of the DIE method closely resembles that of "Great, Now Write an Article About That: The Crescendo Multi-Turn LLM Jailbreak Attack" (published in April), as both methods can be summarized as introducing an unrelated topic before eventually returning to the original malicious intent.
2. In Section 1 (**Introduction**), the authors claim that a key advantage of the DIE method is its ability to bypass LLM defenses (e.g., sensitive word detection). However, this advantage is not convincingly demonstrated in the experiments. The authors may consider adding additional experiments to better illustrate the superiority of the DIE method in this context.
3. Using results from white-box methods as baselines may lead to an unfair comparison. If white-box methods are to be included, the comparison should be made under white-box conditions. Moreover, since DIE is a multi-turn jailbreak method, it would be beneficial to compare it with other multi-turn jailbreak techniques to better contextualize its performance.
4. The paper mentions that the model used is llama-6B, which appears to be a typographical error.

**Questions:**

1. The auxiliary model \( M_a \) plays a crucial role in the experiments, and different configurations of the model may significantly influence the results. The authors claim that \( M_a \) can be any large language model, but the specific configuration of \( M_a \) in the experiments is not clearly explained.
2. Could the authors clarify the differences and innovations in their work compared to "Great, Now Write an Article About That: The Crescendo Multi-Turn LLM Jailbreak Attack"?
3. Could the authors provide more details on the attack effectiveness against defenses such as sensitive word detection?

**Reviewer Confidence:**

4: The reviewer is certain that the evaluation is correct and very familiar with the relevant literature

**Scope:**

4: The work is relevant to the Web and to the track, and is of broad interest to the community

---

### Official Review · Reviewer_bnhv · 2024-12-02

**Novelty:** 4
**Technical Quality:** 3

**Review:**

Summary

This paper proposes a new jailbreak attack method against large language models
(LLMs), which combines two key modules to achieve its objectives. The
Intention-Anchored Malicious Concealment (IMC) module hides harmful intentions
by embedding them behind a generated anchor intention. Meanwhile, the
Intention-Reinforced Malicious Inducement (IMI) module exploits the availability
bias mechanism through a progressive malicious prompting strategy. Experiments
demonstrate the effectiveness of the proposed method.

Strengths

* The studied problem is interesting.

* The proposed attack achieves high attack success rates.

Weaknesses

* The experiments involve only relatively old or small models. To better
demonstrate the generalizability of the findings and methods, it is recommended
to include more recent models, such as LLaMA3.1, Qwen2, and GPT-4o and
Cluade-3.5. Including larger and more advanced models would strengthen the
claims of generalizability.

* There are many existing jailbreak methods [1,2,3] that are not included in the
experiments. Thus, it is not clear if the proposed method outperforms these
existing methods. It is suggested to add more baseline methods to make the
evaluation more comprehensive.

* There are many existing defenses against jailbreak attacks on LLMs, this paper
lacks the detailed experiments and the discussion on the resistance of the proposed
method to existing defenses.


[1] Jiang et al., ArtPrompt: ASCII Art-based Jailbreak Attacks against Aligned LLMs (ArtPrompt). ACL 2024.

[2] Liu et al., Making Them Ask and Answer: Jailbreaking Large Language Models in Few Queries via Disguise and Reconstruction (DRA). USENIX Security 2024.

[3] Chao et al., Jailbreaking Black Box Large Language Models in Twenty Queries (PAIR). arXiv 2023.

**Questions:**

see above

**Reviewer Confidence:**

3: The reviewer is confident but not certain that the evaluation is correct

**Scope:**

3: The work is somewhat relevant to the Web and to the track, and is of narrow interest to a sub-community

---

### Official Review · Reviewer_t1UE · 2024-12-02

**Novelty:** 4
**Technical Quality:** 4

**Review:**

This paper presents a novel jailbreak attack framework called DIE (Dual Intention Escape) targeting large language models (LLMs). The paper identifies two key limitations of existing jailbreak attacks - weak penetrability through LLM defenses and insufficient toxicity in generated outputs. To address these issues, the authors propose a two-module approach inspired by human psychology concepts: Intention-anchored Malicious Concealment (IMC) and Intention-reinforced Malicious Inducement (IMI).This paper conducted extensive experiments in a black-box setting, demonstrating the effectiveness of the framework on models like GPT-3.5-turbo. However, its effectiveness on a broader range of models remains to be verified.

### Strengths:

1. Novel approach. The paper introduces an innovative perspective by incorporating psychology concepts (anchoring effect and availability bias) into jailbreak attack design, which differs from previous technical-focused approaches.
2. Comprehensive framework. The proposed DIE framework addresses both penetrability and toxicity aspects through well-designed modules (IMC and IMI) with clear technical details.
3. Empirical validation. The experimental results demonstrate significant improvements over state-of-the-art methods, with up to 10.6% increase in attack success rates.

### Weaknesses:

1. The validation of the method's effectiveness is insufficient.This method's effectiveness has not been verified on larger-scale models or different types of models.
2. The novelty of the DIE framework awaits validation.

**Questions:**

1. How generalizable is the DIE framework across different types of malicious prompts and LLM architectures? The paper demonstrates success on specific models but broader applicability is unclear.
2. Could the psychological principles used in the attack framework also inform better defense strategies? This seems like a missed opportunity for discussion.

**Reviewer Confidence:**

3: The reviewer is confident but not certain that the evaluation is correct

**Scope:**

4: The work is relevant to the Web and to the track, and is of broad interest to the community